# Impact of Previous Common Human Coronavirus Exposure on SARS-CoV-2-Specific T-Cell and Memory B-Cell Response after mRNA-Based Vaccination

**DOI:** 10.3390/v15030627

**Published:** 2023-02-24

**Authors:** José L. Casado, Pilar Vizcarra, Adrián Martín-Hondarza, Magdalena Blasco, Marta Grandal-Platero, Johannes Haemmerle, Marina Fernández-Escribano, Alejandro Vallejo

**Affiliations:** 1Department of Infectious Diseases, CIBERINFEC (Biomedical Research Center Network in Infectious Diseases, Centro de Investigacion en Red en Enfermedades Infecciosas), Instituto de Salud Carlos III (ISCIII), 28029 Madrid, Spain; 2IRYCIS, Instituto Ramón y Cajal de Investigaciones Sanitarias, 28034 Madrid, Spain; 3Laboratory of Immunovirology, IRYCIS, 28034 Madrid, Spain; 4Department of Occupational Safety and Health, Hospital Universitario Ramón y Cajal, 28034 Madrid, Spain

**Keywords:** SARS-CoV-2, memory B-cells, cross-reactive, coronavirus, mRNA vaccine

## Abstract

Objective: T-cell responses against SARS-CoV-2 are observed in unexposed individuals, attributed to previous common human coronavirus (HCoV) infections. We evaluated the evolution of this T-cell cross-reactive response and the specific memory B-cells (MBCs) after the SARS-CoV-2 mRNA-based vaccination and its impact on incident SARS-CoV-2 infections. Methods: This was a longitudinal study of 149 healthcare workers (HCWs) that included 85 unexposed individuals that were subdivided according to previous T-cell cross-reactivity, who were compared to 64 convalescent HCWs. Changes in specific T-cell response and memory B-cell (MBC) levels were compared at baseline and after two doses of the SARS-CoV-2 mRNA-based vaccine. Results: A cross-reactive T-cell response was found in 59% of unexposed individuals before vaccination. Antibodies against HKU1 positively correlated with OC43 and 229E antibodies. Spike-specific MBCs was scarce in unexposed HCWs regardless of the presence of baseline T-cell cross-reactivity. After vaccination, 92% and 96% of unexposed HCWs with cross-reactive T-cells had CD4+ and CD8+ T-cell responses to the spike protein, respectively. Similar results to that were found in convalescents (83% and 92%, respectively). Contrarily, higher than that which was observed in unexposed individuals without T-cell cross-reactivity showed lower CD4+ and CD8+ T-cell responses (73% in both cases, *p* = 0.03). Nevertheless, previous cross-reactive T-cell response was not associated with higher levels of MBCs after vaccination in unexposed HCWs. During a follow-up of 434 days (IQR, 339–495) after vaccination, 49 HCWs (33%) became infected, with a significant positive correlation between spike-specific MBC levels and isotypes IgG+ and IgA+ after vaccination and a longer time to get infected. Interestingly, T-cell cross-reactivity did not reduce the time to vaccine breakthrough infections. Conclusion: While pre-existing T-cell cross-reactivity enhances the T-cell response after vaccination, it does not increase SARS-CoV-2-specific MBC levels in the absence of previous infection. Overall, the level of specific MBCs determines the time to breakthrough infections, regardless of the presence of T-cell cross-reactivity.

## 1. Introduction

The induction of effective early immune control of SARS-CoV-2 and durable immune memory are critical events to prevent severe disease and to protect upon re-exposure, but there are controversies about the duration of such responses [1,2]. In addition, it has been demonstrated a different kinetic of immune response in SARS-CoV-2 naïve and recovered individuals after vaccination [3,4].

This different immune response in naïve and recovered SARS-CoV-2 individuals could be due to the presence of prior cross-reactive T-cell response against SARS-CoV-2, as cumulative evidence showed cross-reactive T-cell immunity between human coronaviruses (HCoV 229E, NL63, OC43, and HKU1), and SARS-CoV-2 [5,6,7]. Indeed, previous cross-reactivity with common coronavirus has been shown to offer transient protection against infection with SARS-CoV-2 [8,9], alleviating at least disease manifestations from COVID-19.

However, data raise the intriguing possibility that such pre-existing S-reactive T-cells could offer better long-term protection, and that cross-reactive immunity could influence responsiveness to vaccines [10]. To evaluate the hypothesis that a previous common coronavirus could lead to similar boosted spike-specific antibodies and memory B-cell responses after vaccination, we longitudinally analyzed the evolution of adaptative immune response in a cohort of unexposed individuals after mRNA vaccination, according to the presence of T-cell cross-reactivity at baseline and compared to that which was observed in recovered individuals.

## 2. Materials and Methods

This longitudinal study included 149 healthcare workers (HCWs), including 64 convalescent COVID-19 patients, and 85 SARS-CoV-2 unexposed healthy individuals, followed up since March 2020 in the tertiary Ramon y Cajal University Hospital (Madrid, Spain). Before being recruited, HCWs had participated in an internal survey about the presence of antibodies against the N protein of SARS-CoV-2 after the first surge of the disease [11], and after inclusion into the study they were vaccinated with two doses of mRNA-based BNT162b2 (Pfizer BioNTech) COVID-19 vaccine in January–February 2021. Thus, we analyzed three key time points: the internal serological survey (April 2020; COVID-19 IgG/IgM Rapid Test Kit, UNscience Biotechnology, Wuhan, China; and COVID-19-SARS-CoV-2 IgA ELISA, Demeditech, Germany), the inclusion into the study (baseline, October 2020), and 3–4 weeks following the second dose of vaccination (February 2021). This study design allowed us to investigate the kinetics of immune responses following infection and secondary to immunization.

In both unexposed and recovered HCWs, we evaluated (inclusion into the study) the T-cell response to spike (S), membrane (M), and nucleocapsid (N) proteins of SARS-CoV-2 and the rate of different isotypes spike-specific memory B-cell (MBC) isotypes at baseline and after the second dose of mRNA vaccination against SARS-CoV-2. To initially identify those individuals with recent infection, we also examined antibody levels to the spike protein of the common human alpha (229E and NL63) and beta (HKU1 and OC43) HCoVs.

Convalescent or recovered from COVID-19 patients were defined as those individuals with COVID-19 suggestive symptoms and positive nasopharyngeal swab PCR-positive test against SARS-CoV-2. Naïve or unexposed individuals were those individuals without COVID-19 suggestive symptoms, negative nasopharyngeal swab PCR test against SARS-CoV-2 (when performed), and negative for anti-N IgG antibodies in both the internal survey and at the inclusion of the study. This last group was subdivided according to the presence of CD4+ and/or CD8+ T-cell cross-reactive response to SARS-CoV-2 at baseline, regardless of the history of serology against common HCoVs.

At inclusion, all the patients gave written informed consent, and the study was approved by the institutional review boards of our Hospital Ethics Committee (EC162/20) and registered at the clinical trials repository (clinicaltrials.gov first posted 27 May 2020, NCT04402827).

### 2.1. Laboratory Analysis

Peripheral blood mononuclear cells (PBMC) were isolated from EDTA-blood sample by Ficoll-Paque density gradient centrifugation using lymphocyte separation medium (Corning, New York City, NY, USA) and cryopreserved until use. Plasma samples were stored at −80 °C. All the participants (naïve and convalescents) were tested for anti-N SARS-CoV-2 IgG antibodies (COVID-19-SARS-CoV-2 IgG ELISA, Demeditech Diagnostics GmbH, Kiel, Germany) at inclusion and after a median of 17 days after the second dose of vaccination to confirm the serologic status regardless of the antibody production following the vaccine. The results were recorded as relative units per milliliter (U/mL), with a threshold of 11 U/mL. They were also tested for anti-spike IgG antibodies (SARS-CoV-2 IgG II Quant, Abbott, Maidenhead, UK) with a threshold of 50 arbitrary units per milliliter (AU/mL). We tested IgG antibodies to the four common human coronaviruses that are responsible for the seasonal upper respiratory tract infections (Recombivirus human anti-HCoV HKU1, NL63, 229E, and OC43 S1 IgG ELISA kits, Alpha Diagnostic, San Antonio, TX, USA), but without establishing a cut-off value since these antibodies are detected in most of the individuals and are ubiquitous with winter seasonality.

### 2.2. Determination of SARS-CoV-2 Neutralizing Antibody

SARS-CoV-2 neutralizing antibodies were quantified using a competitive inhibition enzyme immunoassay technique (Human Novel Coronavirus (SARS-CoV-2) Neutralizing Antibody ELISA Kit, MyBioSource, San Diego, CA, USA) following the manufacturer’s instructions. Plate wells were pre-coated with SARS-CoV-2 RBD and horseradish peroxidase-conjugated ACE2 was added with the sample. The competitive inhibition reaction was launched between HRP-ACE2 and SARS-CoV-2 neutralizing antibodies in samples. A substrate solution was added to the wells and the color developed opposite to the amount of SARS-CoV-2 neutralizing antibody in the sample. Optical densities greater than half the optical density for the blank were considered negative. The results were recorded as ng/mL.

### 2.3. Determination of SARS-CoV-2 Spike-Specific Memory B-Cells

SARS-CoV-2-specific MBC detection was performed by binding the recombinant spike protein to the respective antigen-specific B-cell receptor (BCR) on circulating B-cells (SARS-CoV-2 spike B-cell analysis kit, Miltenyi Biotec, Germany) by multiparametric flow cytometry (Figure 1A for cytometry strategy).

Tetramers that formed from recombinant SARS-CoV-2 Spike-Prot (HEK)-Biotin with Streptavidin, PE, and PE-Vio770, respectively, were used according to the manufacturer´s instructions. This quantitative and qualitative analysis of specific MBC and isotypes IgG+, IgM+, and IgA+, was performed by single-cell flow cytometry with fluorochrome-conjugated antibodies, and the 7-AAD for the exclusion of dead and apoptotic cells using a minimum of 5 × 10^6^ PBMCs for each analysis. The results are recorded as a percentage among total memory B-cells and isotypes among specific MBCs.

### 2.4. Determination of SARS-CoV-2 Spike-Specific T-Cells

Specific CD4+ and CD8+ T-cell responses were analyzed by intracellular cytokine staining using multiparametric flow cytometry. Briefly, SARS-CoV-2-specific T-cells were measured using in vitro stimulation with SARS-CoV-2 peptide pools of viral proteins encompassing the spike (S), membrane (M), and nucleocapsid (N) protein followed by quantitation of CD4+ and CD8+ T-cell-specific interferon (IFN)-γ, using peripheral blood mononuclear cell (PBMC) samples from all subjects. A result that was 2-fold higher than the negative control (unstimulated) was considered positive. The complete flow cytometry strategy is shown in Figure 1B.

### 2.5. Statistical Analysis

Continuous variables were expressed as the median and interquartile range (IQ_25–75_) and categorical variables by frequencies and proportions. The Mann–Whitney *U* test (non-parametric) for independent samples was used to compare continuous variables. The Wilcoxon signed-rank test was used to compare paired samples to analyze the evolution of the measurements after vaccination. Spearman’s rank correlation coefficient was used to measure the association between two variables. Differences between categorical variables were evaluated using contingency tables (Chi-square distribution). Statistical significance was defined as two-sided *p*-values below 0.05.

## 3. Results

We included 149 HCWs with no self-reported chronic health conditions or immunosuppression who were analyzed for T-cell cross-reactive response at baseline and after receiving two doses of SARS-CoV-2 mRNA vaccine Pfizer BNT162b2. Demographic and baseline information is described in Table 1. Of them, 85 individuals had no previous COVID-19 disease, as confirmed by no positive PCR or anti-N positive serology at the survey and inclusion in the study. The control group was composed of 64 recovered individuals who had a prior SARS-CoV-2 infection with a median time of 198 days (interquartile range, IQR, 179–216) before the evaluation of the immune system.

Data are expressed as the median and interquartile range, and percentage. ANOVA for continuous and chi-square for categorical variables for statistical differences between cross-reactive and convalescent individuals.

At inclusion, HCoV HKU1 IgG antibodies correlated with HCoV 229E (r = 0.322; *p* = 0.045) and OC43 (r = 0.363; *p* = 0.021) IgG antibodies especially in unexposed individuals (r = 0.525 and r = 0.418; *p* = 0.025 and *p* = 0.034, respectively; Figure 2A).

Of note, HCoV antibody levels did not correlate with SARS-CoV-2 anti-S or anti-N IgA or IgM antibodies. Nevertheless, OC43 antibody levels significantly correlated with SARS-CoV-2 anti-N IgG (r = 0.463; *p* = 0.05) and neutralizing (r = 0.857; *p* = 0.014) antibodies at inclusion in convalescent individuals (Figure 2B). Also, only CD4+ T-cell against S protein tended to be associated with HCoV OC43 antibodies (r = 0.286, *p* = 0.090). Finally, neither spike-specific MBCs nor the MBC isotypes showed any association with the different HCoV antibodies.

Moreover, convalescent subjects showed SARS-CoV-2-specific MBCs in 81% of cases at a lower magnitude of isotypes IgM and IgA (Figure 3). As mentioned, SARS-CoV-2-naïve individuals had no specific IgG antibodies for either full-length spike protein or N protein, suggesting the absence of previous infection. SARS-CoV-2-specific MBCs were found in only two cases (5%) in unexposed subjects with no T-cell cross-reactivity and with a very low levels of response (Figure 3).

Initially, 59 out of 85 (69%) unexposed HCWs showed a T-cell cross-reactive response against structural proteins of SARS-CoV-2, including CD4+ response towards S (59%), M (52%) and N (52%) proteins, and CD8+ response towards S (49%), M (53%), or/and N (49%) proteins (Figure 4). Of note, CD4+ T-cell responses to the three SARS-CoV-2 proteins was present in 41%, 36%, and 36% in unexposed subjects, respectively.

### 3.1. Association between Specific T-Cells and MBCs after SARS-CoV-2 Vaccine according to Cross-Reactivity

Specific MBCs increased in all individuals after the second dose of the vaccine (Figure 3). However, the level of spike-specific MBCs was significantly lower in the unexposed subjects compared to convalescents. Neutralizing and anti-S IgG antibodies were detected in all individuals, and again the magnitude of response was associated with previous infection.

After two doses of the mRNA vaccination, 92% and 96% of participants with cross-reactive T-cells had CD4+ and CD8+ T-cells against S, such as convalescent (83% and 92%). These frequencies were higher than that which was observed in unexposed individuals without cross-reactivity (73% in both cases; Figure 4). Indeed, cross-reactivity significantly increased the magnitude of CD8+ T-cell response (*p* = 0.03). In the global population, previous exposure to HCoV HKU1 significantly increased the CD8+ T-cell response against N protein (r = 0.597; *p* = 0.019) and CD4+ T-cell response against M protein (r = 0.481; *p* = 0.049). Nevertheless, MBC levels after the second dose of the mRNA vaccination were not associated with previous HCoV serology. Nevertheless, a higher level of IgM+ MBCs was observed in convalescent individuals, with a trend for IgA+ MBC levels (r = 0.583; *p* = 0.099) (Figure 5A,B).

### 3.2. Incident Infections after mRNA Vaccination

There was a significant inverse correlation between spike-specific MBC (r = −0.307; *p* = 0.027), especially for IgA isotype (r = −0.279, *p* = 0.045), neutralizing antibodies (r = −0.265; *p* = 0.44), and anti-S IgG (r = −0.288; *p* = 0.028) and IgA (r = −0.32; *p* = 0.031) antibodies with the time from the second dose of vaccination, despite the short time between vaccine administration and analysis (median, 27 days, IQR, 25–29).

As a result, during a median follow-up of 434 days (IQR, 339–495) after the two-doses of the vaccine regimen, 49 individuals (33%) became infected, not requiring hospitalization, without differences according to cross-reactivity in frequency (34% vs. 35%) or time to incident infection. Interestingly, there was an overall significant direct correlation between spike-specific MBC levels after vaccination and the time for SARS-CoV-2 infection, especially for the total specific MBC and IgA+/IgG+ MBC isotypes. Notably, this correlation was also observed for IgA antibodies (r = 0.563; *p* = 0.015).

## 4. Discussion

Our study confirms that pre-existing S-reactive T-cells represent cross-reactive clones secondary to previous infections with endemic HCoV. In addition, we showed that previous HCoV exposure primed an early neutralizing antibodies response, and partly of IgG and IgA, in SARS-CoV-2 convalescent individuals. We also showed here that the presence of cross-reactivity elicits a better CD8+ T-cell response in unexposed individuals, but it does not elicit an increase of IgA+/IgG+ MBC isotypes after vaccination in convalescents, independently of previous HCoV exposure.

Although all individuals that were included in this and other studies were seropositive to HCoVs at different levels, consistent with the endemic nature of these viruses [12,13], the rate of T-cell cross-reactivity varied in the overall population [5]. It has been described, in agreement with our results, that antibodies against HCoVs did not correlate with cross-reactive CD4+ T-cells because they have not been recently generated, as observed with early CD4+ T-cell responses following yellow fever vaccination [14]. Nevertheless, we observed that an important number of unexposed individuals had T-cell responses, similar in magnitude to that observed in convalescents. Indeed, the rate and magnitude of response to M and N proteins were similar in convalescents and unexposed subjects confirming the role of the homology between proteins of different coronaviruses.

Furthermore, it has been described a hierarchy of SARS-CoV-2-specific CD4+ T-cell targets, with the majority of the CD4+ T-cell response in COVID-19 cases directed against highly expressed SARS-CoV-2 spike, and less against M and N proteins [5]. Thus, a different epitope presentation could be expected in cases with cross-reactivity. However, cross-reactive SARS-CoV-2-specific CD8^+^ T-cells directed against epitopes that are highly conserved among HCoVs are now well described, with pre-existing T-cells frequently targeting essential viral proteins [15]. Moreover, cross-reactive T-cells showed a predominance of response to M protein in our study (52% of convalescent and 36% unexposed), in the latter case probably explained due to 90% structural identity with that of other coronaviruses [16]. In addition, we demonstrated that SARS-CoV-2 mRNA vaccines, encoding the full-length S protein, might trigger the cellular response to other viral proteins in individuals with previous T-cell cross-reactivity, although of uncertain clinical significance [17].

Importantly, few studies have evaluated the role of humoral and neutralizing response, and the level of MBC isotypes after SARS-CoV-2 vaccination according to cross-reactivity. Loos et al. [18] reported some positive relationships between the IgG antibody response to the common HCoVs and SARS-CoV-2 but suggested that cross-reactive immunity plays a limited role in shaping SARS-CoV-2 humoral immune responses. In a similar study that was performed in convalescent individuals, specific MBCs after SARS-CoV-2 infection showed some clones that were cross-reactive to the S proteins of HCoVs OC43 and HKU1, or both [19]. Thus, as we also have shown, in the context of COVID-19, prior HCoV exposure could help to activate neutralizing, IgG, and IgA antibodies by antigen recognition. Furthermore, we found a significant correlation between IgM+ MBCs with a trend for IgA+ MBC after vaccination in convalescent individuals. However, it seems to be different in unexposed subjects. The relation between cross-reactive T-cells and humoral immune evolution after SARS-CoV-2 vaccination was found to be associated with a lower serologic response than that which was observed in convalescents [20,21]. Data from human coronaviruses suggest the possibility that substantive adaptive immune responses can fail to occur [22]. In parallel with our findings, other studies have described that pre-existing cross-reactive memory T-cells found in some uninfected HCWs may lead to abortive seronegative infections [23]. Besides, since pre-existing SARS-CoV-2 T-cell responses also protect from disease, the fact that their presence was not associated with breakthrough infection is not unexpected and not in contrast with abortive infections.

As a possible explanation, the expansion of pre-existing memory T-cells predates antibody induction after mRNA vaccination [24,25], similar to that which was observed in some individuals with mild SARS-CoV-2 infection [26]. Nevertheless, cross-reactive T-cells were associated with cross-protection or with attenuated symptoms (protection from disease). CD8^+^ T-cells that act against seasonal coronaviruses may be stimulated by vaccination and they could be skewed toward SARS-CoV-2 with a partial protective role, in agreement with our results [27].

We acknowledge the limitations of our study. Only individuals with a recent vaccination schedule were studied, hence studying MBC levels in patients without SARS-CoV-2 infection should be considered in subsequent studies to evaluate longer follow-up periods to decipher potential intra- and inter-variability. The study was limited to the analysis after the second vaccine dose and not extended to the third vaccine dose, especially for the role of new infections with different variants of concern. Indeed, we observed breakthrough infections with a predominance of Delta (during 2021 in our milieu) and Omicron variants (during 2022). Similar to most studies of adaptive immune responses in humans, our analysis was limited to the peripheral blood even though MBCs were likely generated in secondary lymphoid organs. Finally, asymptomatic infections and misclassification of cross-reactivity in unexposed individuals were certainly possible. Moreover, breakthrough infections were only defined as the presence of symptoms and positive nasopharyngeal swab PCR test, hence, we were unable to diagnose asymptomatic infections.

In conclusion, we demonstrated the role of HCoVs in establishing cross-reactive T-cell response against SARS-CoV-2, and that T-cell response is further stimulated after two doses of mRNA vaccination. On the other hand, cross-reactivity was not associated with higher MBC levels after vaccination in naïve SARS-CoV-2 individuals. In any case, this specific MBC level is associated with a longer time to breakthrough infections even in the context of new variants of concern, pointing out the need for additional vaccine doses for establishing durable immune protection from disease.

## Figures and Tables

**Figure 1 viruses-15-00627-f001:**
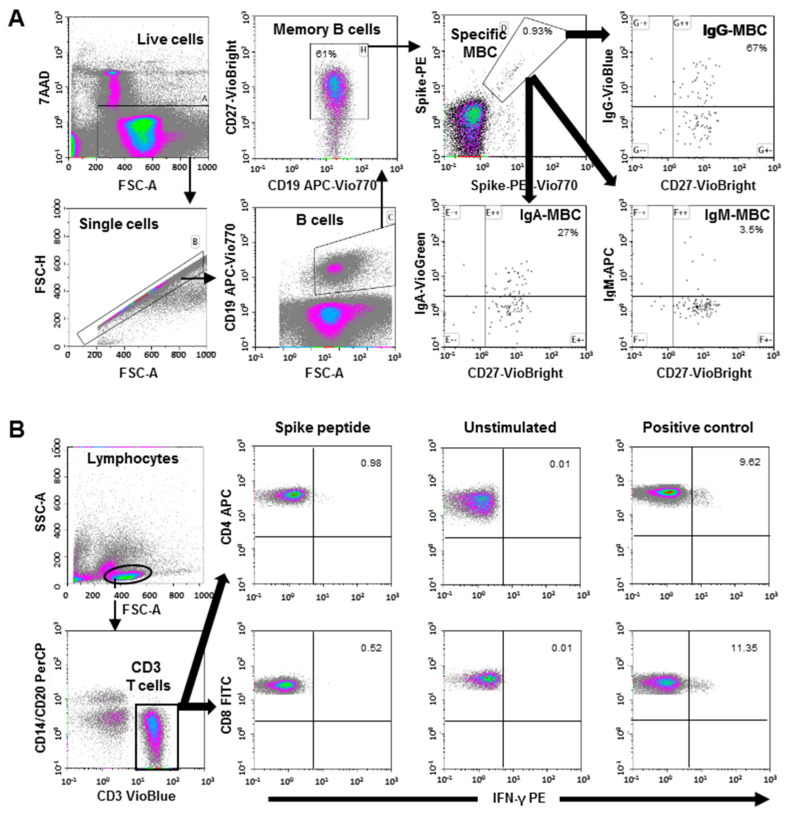
(**A**) Flow cytometry strategy for the quantification of SARS-CoV-2-specific memory B-cells (MBC). Viable cells (FSC-A/7AAA plot) were plotted with FSC-H/FSC-A parameters to exclude doublets. Single cells were then gated using CD19-APC-Vio770 and CD27-Vio-Bright-FITC to identify memory B-cells. Spike-specific B-cells were then identified with a double staining with the two spike-tetramer on the diagonal of the dot plot. Finally, the use of IgG-VioBlue, IgA-VioGreen, and IgM-APC were used to quantify each specific isotype of spike-specific memory B-cells. The results are recorded as percentage among the total memory B-cells and isotypes among specific memory B-cells. (**B**) Flow cytometry gating strategy for the quantification of SARS-CoV-2 spike-specific T-cells. After stimulation with the spike peptide, staining of the cells was carried out with the following fluorochrome-conjugated antibodies: CD3-VioBlue, CD4-APC, CD8-FITC, CD14-PerCP, CD20-PerCP, IFN-γ-PE, and FcR blocking reagent. To exclude dead cells, viability 405/520 fixable dye staining was added for the last 10 min of incubation. The samples were measured and analyzed by flow cytometry on a MACSQuant Analyzer 10 using MACSQuantify software. At least 10^5^ cells were analyzed and gated with the following strategy: Single (FSC-A/FSC-H dot plot) and live cells were first selected. Cell debris, monocytes, and B-cells were excluded from the analysis with CD14-and CD20-PerCP antibodies. Then, lymphocytes were selected with an FSC-A/SSC-A dot plot, and CD3 T-cells were gated. IFN-γ expression was finally analyzed separately for CD4+ and CD8+ T-cells. A representative sample of negative control (without stimulation), positive control (stimulated with SEB), and with SARS-CoV-2 spike peptide are shown.

**Figure 2 viruses-15-00627-f002:**
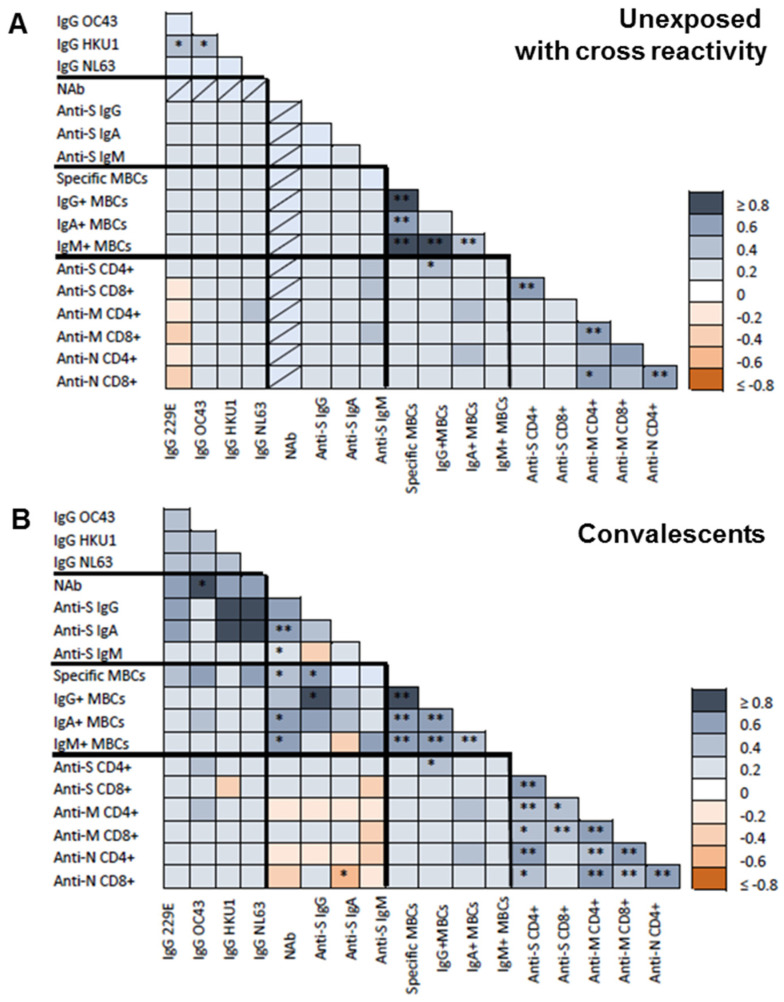
Baseline correlations between antibodies against common human coronavirus (HCoVs), SARS-CoV-2 spike-specific (IgG, IgA, and IgM) antibodies, neutralizing antibodies (NAB), specific memory B-cells (MBCs) and IgG+, IgA+, and IgM+ MBCs isotypes, CD4+ and CD8+ T-cell responses to SARS-CoV-2 spike (S), membrane (M), and nucleocapsid (N) proteins. (**A**) Spearman test heatmap analysis of the different variables in unexposed subjects with cross-reactivity. (**B**) Spearman test heatmap of the different variables in convalescent individuals. The grey color represents a positive correlation and the red color a negative correlation. The intensity of the color indicates the R^2^ coefficient. Statistically significant when *p* < 0.05. * = *p* < 0.05, and ** = *p* < 0.01.

**Figure 3 viruses-15-00627-f003:**
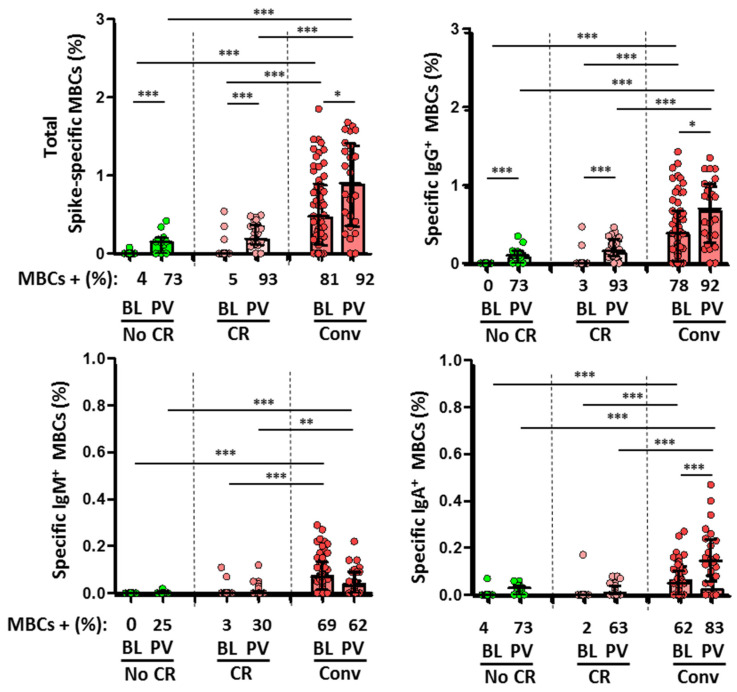
Comparison of specific memory B-cells (MBCs, **top left**) and IgG+ MBC (**top right**), IgM+ MBCs (**bottom left**), and IgA+ MBCs (**bottom right**) isotypes between the different subgroups at baseline (BL) and after two doses of vaccine (PV, post-vaccination). No CR (unexposed without cross-reactivity, green dots and bars), CR (unexposed with cross-reactivity, rose dots and bars), and Conv (convalescent, red dots and bars). Frequencies are indicated at the X axis. Statistically significant when *p* < 0.05. * = *p* < 0.05, ** = *p* < 0.01, and *** = *p* < 0.001.

**Figure 4 viruses-15-00627-f004:**
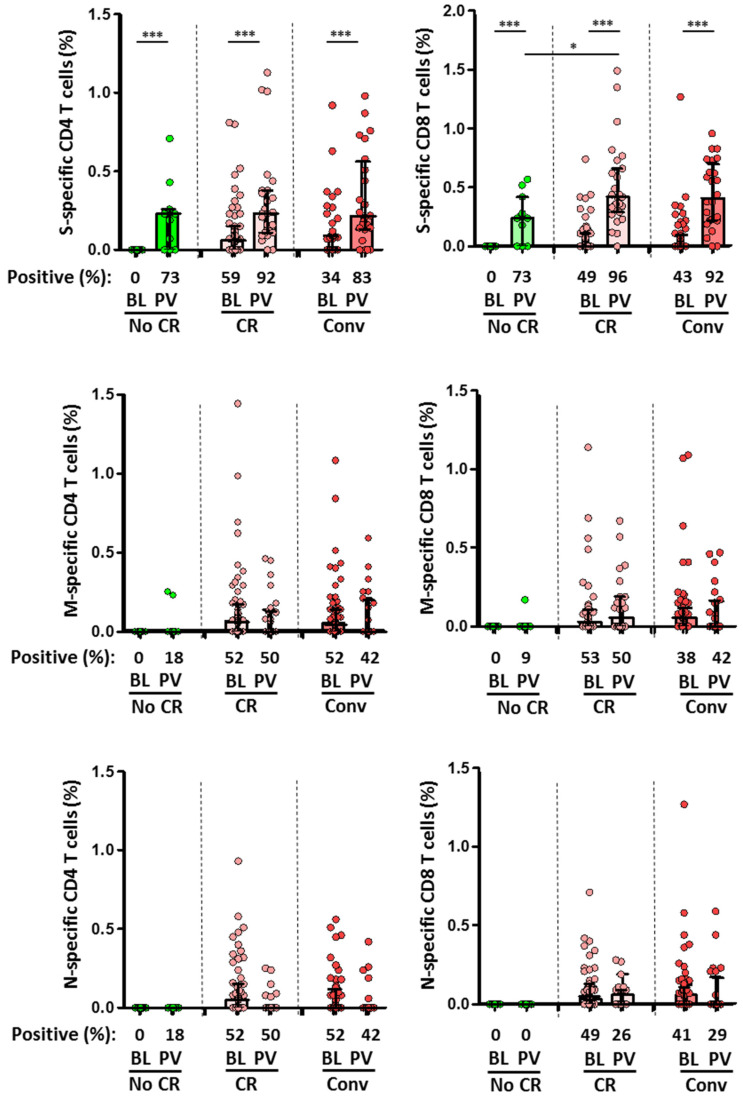
Comparison of the magnitude of CD4+ (**left**) and CD8+ (**right**) T-cell responses against SARS-CoV-2 S, M, and N proteins between the different subgroups at baseline (BL) and after two doses of vaccine (PV, post-vaccination). No CR (unexposed without cross-reactivity, green dots and bars), CR (unexposed with cross-reactivity, rose dots and bars), and Conv (convalescent, red dots and bars). Frequencies are indicated at the X axis. Statistically significant when *p* < 0.05. * = *p* < 0.05, and *** = *p* < 0.001.

**Figure 5 viruses-15-00627-f005:**
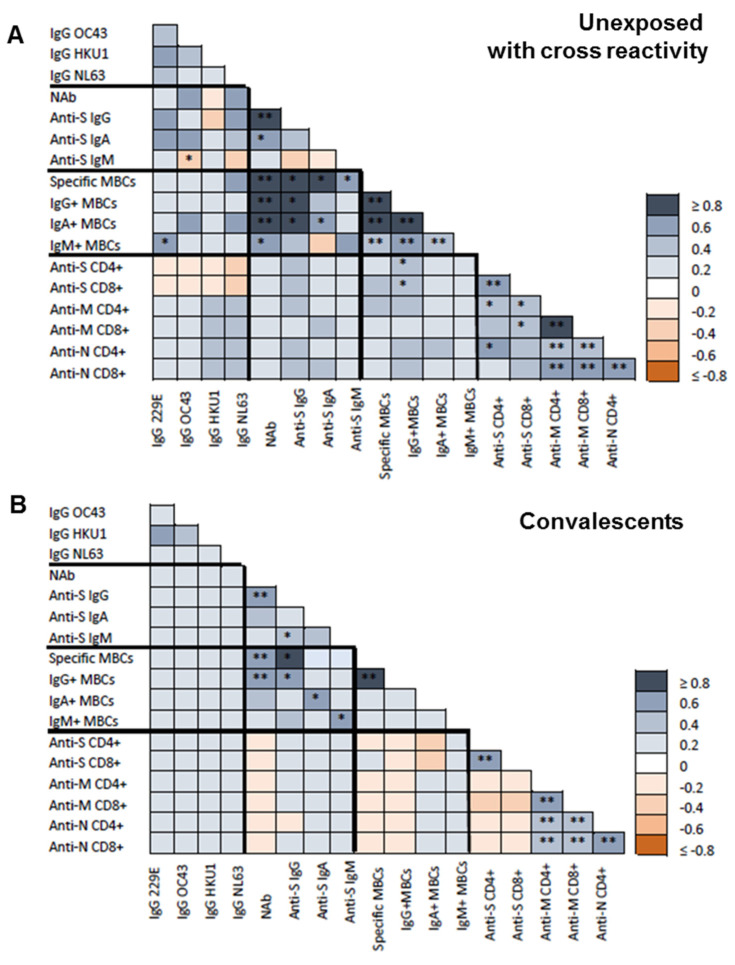
Post-vaccination correlations between antibodies against common human coronavirus (HCoVs), SARS-CoV-2 spike-specific (IgG, IgA, and IgM) antibodies, neutralizing antibodies (NAB), specific memory B-cells (MBCs) and IgG+, IgA+, and IgM+ MBCs isotypes, CD4+ and CD8+ T-cell responses to SARS-CoV-2 spike (S), membrane (M), and nucleocapsid (N) proteins. (**A**), Spearman test heatmap analysis of the different variables in unexposed subjects with cross-reactivity. (**B**), Spearman test heatmap of the different variables in convalescent individuals. The grey color represents a positive correlation and the red color a negative correlation. The intensity of the color indicates the R^2^ coefficient. Statistically significant when *p* < 0.05. * = *p* < 0.05, and ** = *p* < 0.01.

**Table 1 viruses-15-00627-t001:** Characteristics of the individuals that were included in this study.

	Unexposed without Cross-Reactivity (No CR, N = 26)	Unexposed with Cross-Reactivity (CR, N = 59)		
			Convalescent (Conv, N = 64)	ANOVA *p*
Age (years)	44 (26–63)	46 (24–66)	41 (26–67)	0.184
Sex (Female)	20 (77%)	41 (70%)	41 (64%)	0.36 0.392
Body Mass Index	22.5 (20–27)	23.6 (21–26.2)	23.6 (22–26.8)	
Obesity (>30)	50 (13%)	4 (7%)	8 (13%)	
Smoking	3 (12%)	19 (32%)	25 (39%)	0.005
Diabetes mellitus	1 (4%)	3 (5%)	0	0.098
Hypertension	0	6 (10%)	6 (9%)	0.781
Days from COVID-19 to inclusion	-	-	198 (179–216)	-
Serology at inclusion:				
Anti-N IgG positive (N)	0	0	37 (58%)	<0.001
IgG antibody titer (AU/mL)	4.6 (4–6.3)	4.66 (3.7–6.1)	10.1 (5.3–19.8)	<0.001
Neutralizing antibodies (ng/mL)	-	-	1692 (1004–2562)	
Anti-S T-cell response:				
CD4+ T-cell positive (N)	0	35 (59%)	33 (52%)	0.792
CD8+ T-cell positive (N)	0	31 (53%)	28 (44%)	0.570

## Data Availability

The raw data supporting the conclusions of this article will be made available by the authors, without undue reservation.

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
