# Peer review of "Impact of Previous Common Human Coronavirus Exposure on SARS-CoV-2-Specific T-Cell and Memory B-Cell Response after mRNA-Based Vaccination"

_viruses, 2023, doi:10.3390/v15030627_

Round 1
Reviewer 1 Report
This work analyzed the impact of previous human Coronavirus exposure on SARS-CoV-2 specific T and B cell memory. The authors suggested that presence of T cell cross-reactivity against " common human Coronavirus " influences the expansion of Spike-specific CD4 and CD8 but not of B cells. They also show that level of antibodies influences the time of breakthrough infection, while T cell response is not.
The data are performed in a large group of healthy individuals and they are technically sounds.
Major Concerns:
I have major reservations about the classification of unexposed SARS-CoV2 individuals based exclusively in absence of symptoms and absence of anti-N antibodies. It is well know that asymptomatic SARS-CoV-2 infected individuals are often not producing antibodies ( see Sekine, T. et al. Robust T Cell Immunity in Convalescent Individuals with Asymptomatic or Mild COVID-19. Cell 183, 158-168.e14 (2020) and Le Bert, N. . et al. Highly functional virus-specific cellular immune response in asymptomatic SARS-CoV-2 infection. J Exp Med 218, e20202617 (2021).). In addition it is also well established that anti-N antibodies can revert to negativity very often ( Group, T. C.-19 C. R. P. S. et al. Duration of SARS-CoV-2 sero-positivity in a large longitudinal sero-surveillance cohort: the COVID-19 Community Research Partnership. Bmc Infect Dis 21, 889 (2021).). It is also well established that T cell cross-reactivity against common coronavirus is characterised by presence of single T cell specialities and not by a T cell response specific against different proteins of SARS-CoV-2 ( NP, M and Spike) see (Ogbe, A. et al. T cell assays differentiate clinical and subclinical SARS-CoV-2 infections from cross-reactive antiviral responses. Nat Commun 12, 2055 (2021).) The presence of N, M and S specific CD4 and CD8 T cells in the individuals labelled as " SARS-CoV-2 unexposed" is extremely problematic. It is more likely that these individuals were infected asymptomatically by SARS-CoV-2 an cannot be defined as " unexposed". Certainly the absence of antibodies against N or the absence of symptoms cannot exclude previous infection. The authors should analyse better their T cell response and perhaps include in their unexposed individuals only individuals with a single T cell specificities. They should also test T cell response in PBMC collected before 2020 and see whether their rate of cross-reactivity ( 60%) could be demonstrated also before 2020. Common coronaviruses were present even before 2020 so they might be able to perhaps challenge the current idea that presence of multiple SARS-CoV-2-specific T cell response is an indication of previous SARS-CoV-2 infection. It is clear that without a clear distinction between unexposed and exposed all the data should be changed.
2) It is not clear why T cell specific for M or N or Spike should protect from infection. SARS-CoV-2 specific T cells protect from disease thus the fact that T cell response against Spike was not correlate with time of breakthrough infection is not unexpected and not in contrast with data related to abortive infection were T cell response against polymerase was tested. There is a clear distinction between protection from infection and protection from disease and the authors should clearly state such difference and discuss their findings accordingly.
Author Response
Thank you very much for your review.
This work analyzed the impact of previous human Coronavirus exposure on SARS-CoV-2 specific T and B cell memory. The authors suggested that presence of T cell cross-reactivity against "common human Coronavirus " influences the expansion of Spike-specific CD4 and CD8 but not of B cells. They also show that level of antibodies influences the time of breakthrough infection, while T cell response is not.
The data are performed in a large group of healthy individuals and they are technically sounds.
Major Concerns:
I have major reservations about the classification of unexposed SARS-CoV2 individuals based exclusively in absence of symptoms and absence of anti-N antibodies. It is well know that asymptomatic SARS-CoV-2 infected individuals are often not producing antibodies (see Sekine, T. et al. Robust T Cell Immunity in Convalescent Individuals with Asymptomatic or Mild COVID-19. Cell 183, 158-168.e14 (2020) and Le Bert, N. et al. Highly functional virus-specific cellular immune response in asymptomatic SARS-CoV-2 infection. J Exp Med 218, e20202617 (2021).). In addition it is also well established that anti-N antibodies can revert to negativity very often (Group, T. C.-19 C. R. P. S. et al. Duration of SARS-CoV-2 sero-positivity in a large longitudinal sero-surveillance cohort: the COVID-19 Community Research Partnership. Bmc Infect Dis 21, 889 (2021).). It is also well established that T cell cross-reactivity against common coronavirus is characterised by presence of single T cell specialities and not by a T cell response specific against different proteins of SARS-CoV-2 ( NP, M and Spike) see (Ogbe, A. et al. T cell assays differentiate clinical and subclinical SARS-CoV-2 infections from cross-reactive antiviral responses. Nat Commun 12, 2055 (2021).) The presence of N, M and S specific CD4 and CD8 T cells in the individuals labelled as "SARS-CoV-2 unexposed" is extremely problematic. It is more likely that these individuals were infected asymptomatically by SARS-CoV-2 an cannot be defined as "unexposed". Certainly the absence of antibodies against N or the absence of symptoms cannot exclude previous infection. The authors should analyse better their T cell response and perhaps include in their unexposed individuals only individuals with a single T cell specificities. They should also test T cell response in PBMC collected before 2020 and see whether their rate of cross-reactivity (60%) could be demonstrated also before 2020. Common coronaviruses were present even before 2020 so they might be able to perhaps challenge the current idea that presence of multiple SARS-CoV-2-specific T cell response is an indication of previous SARS-CoV-2 infection. It is clear that without a clear distinction between unexposed and exposed all the data should be changed.
Response: This is a very good concern. In fact, we do agree with the referee and include this fact in the limitations. On the other hand, we would like to consider some definitions. The term “unexposed” refers to the fact that those individuals did not have a high probability of close contact with infected individuals. In contrast to the group of convalescent HCWs who had very close contact with infected patients from the beginning of the pandemic (end of February- March 2020, in Spain).
Pre-existing T-cell response to SARS-CoV-2 has been reported in 30–60% of unexposed individuals (Grifoni A, Weiskopf D, Ramirez SI, Mateus J, Dan JM, Moderbacher CR, et al. Targets of T Cell Responses to SARS-CoV-2 Coronavirus in Humans with COVID-19 Disease and Unexposed Individuals. Cell 2020; 181(7): 1489–1501.e15.: Le Bert N, Tan AT, Kunasegaran K, Tham CYL, Hafezi M, Chia A, et al. SARS-CoV-2-specific T cell immunity in cases of COVID-19 and SARS, and uninfected controls. Nature 2020; 584 (7821): 457–62.). Grifoni et al analyzed T cell responses of unexposed individuals recruited before 2019 and found responses to spike and non-spike proteins. Le Bert et al, included as unexposed donors either sampled before 2019 or serologically negative for both SARS-CoV-2 neutralizing antibodies and SARS-CoV-2 N antibodies. They found individuals with T cell response to N and non-structural proteins. Hence, it is not rare to find T cell responses to more than one protein in the same individual.
We found unexposed individuals with pre-existing T cell responses and we are aware that asymptomatic infections and misclassification of cross-reactivity in these individuals were certainly possible. Nevertheless, we are pretty confident of our classification since these individuals were tested for specific antibodies around one month after the initiation of the pandemic (April 2020). This would rule out individuals who may have developed antibodies but could not lose them in such a short period of time. Yet, it does not rule out an individual with an asymptomatic infection without the development of antibodies. Moreover, these individuals did not show specific MBC.
We found 29% individuals with a single pre-existing T cell specificity and 49% with two specificities. Only 22% individuals had reactivity to the three peptides. Unfortunately, we do not have stored cells from these individuals before 2020 to perform the analysis of the specific T cell responses.
2) It is not clear why T cell specific for M or N or Spike should protect from infection. SARS-CoV-2 specific T cells protect from disease thus the fact that T cell response against Spike was not correlate with time of breakthrough infection is not unexpected and not in contrast with data related to abortive infection were T cell response against polymerase was tested. There is a clear distinction between protection from infection and protection from disease and the authors should clearly state such difference and discuss their findings accordingly.
Response: We agree with that definition. We have modified the discussion according to that distinction.
Reviewer 2 Report
The Manuscript by Casado and co-authors revealed the intricate effects of pre-exposure to seasonal CoV on the humoral and cell-mediated immunity. This work is definitely of interest for immunologists.
I have only minor remarks:
p.2 "positive nasopharyngeal swab PCR positive test against SARS-CoV-2" - might be some typo
Fig. 1a Gating strategy shows that only CD27-positive cells are gated. Yet, at the IgA/IgG/IgM vs CD27 plots CD27-negative cells are visible. Why?
Discussion "Also, for the first time, we demonstrated that the presence of cross-reactivity elicits a better CD8+ T cell in unexposed individuals..." - consider revising
Author Response
Thank you very much for your comments.
The Manuscript by Casado and co-authors revealed the intricate effects of pre-exposure to seasonal CoV on the humoral and cell-mediated immunity. This work is definitely of interest for immunologists.
I have only minor remarks:
p.2 "positive nasopharyngeal swab PCR positive test against SARS-CoV-2" - might be some typo
Response: Fixed
Fig. 1a Gating strategy shows that only CD27-positive cells are gated. Yet, at the IgA/IgG/IgM vs CD27 plots CD27-negative cells are visible. Why?
Response: It was an error. Now it is fixed
Discussion "Also, for the first time, we demonstrated that the presence of cross-reactivity elicits a better CD8+ T cell in unexposed individuals..." - consider revising
Response: Changed to “"Also, we showed that the presence of cross-reactivity elicits a better CD8+ T cell in unexposed individuals..."